# Academic Performance according to School Coexistence Indices in Students from Public Schools in the South of Chile

**DOI:** 10.3390/bs13020154

**Published:** 2023-02-10

**Authors:** Caterin Diaz-Vargas, Andrea Tapia-Figueroa, Jacqueline Valdebenito-Villalobos, María Aurora Gutiérrez-Echavarria, Carmen Claudia Acuña-Zuñiga, Jeanette Parra, Ana María Arias, Lilian Castro-Durán, Yasna Chávez-Castillo, Carlos Cristi-Montero, Rafael Zapata-Lamana, María Antonia Parra-Rizo, Igor Cigarroa

**Affiliations:** 1Escuela de Educación, Universidad de Concepción, Los Ángeles 4440000, Chile; 2Facultad de Educación, Universidad de Concepción, Concepción 40300000, Chile; 3Facultad Ciencias Sociales, Estudiante de Doctorado en Psicología, Universidad de Concepción, Concepción 4030000, Chile; 4IRyS Group, Physical Education School, Pontificia Universidad Católica de Valparaíso, Valparaíso 2581967, Chile; 5Faculty of Health Sciences, Valencian International University (VIU), 46002 Valencia, Spain; 6Department of Health Psychology, Faculty of Social and Health Sciences, Campus of Elche, Miguel Hernandez University (UMH), 03202 Elche, Spain; 7Escuela de Kinesiología, Facultad de Salud, Universidad Santo Tomás, Los Ángeles 4440000, Chile

**Keywords:** academic performance, accumulated final grades, Chile, grade point average, interpersonal relationships, school coexistence

## Abstract

School coexistence is a fundamental aspect for good academic performance. The objective of the study was to identify school coexistence indices, and to analyze differences in academic performance according to these indices in students from public schools in the province of Biobío, Chile. This cross-sectional study involved 730 children (53.8% boys; 12 ± 1.2 years). School coexistence indices as a quality of interpersonal relationships between school bodies, the perception of violence and aggressiveness from a gender perspective, and the perception of levels of safety and unsafety in different school areas as well as academic performance through accumulated final grades (AFG) and grade point averages (GPA) were measured. A total of 40.9% and 41.3% of schoolchildren agreed or strongly agreed that stronger students are violent toward weaker students and boys are violent toward one another, respectively. The school areas most classified as unsafe or very unsafe were the restrooms (20.4%), followed by the playgrounds (10%), and the gym and fields (9.5%). Schoolchildren who classified the relationships within the school bodies as bad, or very bad, presented significantly lower AFG in subjects such as math, language (Spanish), and physical education and health as well as GPA. In the same line, those who perceived greater violence and aggressiveness among peers and higher insecurity in different school areas presented significantly poorer academic performance. In conclusion, students perceived violence and aggressiveness among themselves, and the school areas perceived as unsafe were identified. Furthermore, students who perceived poorer school coexistence indices presented a weaker academic performance.

## 1. Introduction

The factors influencing academic performance are multiple and diverse, ranging from personal to sociocultural aspects [1,2,3]. In this context, school coexistence is defined as the multifarious forms of interaction between the members of a school body [4]. Moreover, evidence points out that poor school coexistence could negatively affect academic performance [5]. Specifically, inappropriate student behavior in the classroom tends to hinder and impede the normal development of a class since the teacher and students are disturbed, affecting group performance, and generating a tense class environment, which produces poor interpersonal relationships among the people involved [6].

As stated by [7], the students’ perceptions of their schools’ coexistence can greatly affect their general academic performance. One of the possible explanations for this phenomenon is that conflict, which is relatively common among peers, psychological and verbal abuse, and the absence of rules that are known and respected by everybody in the classroom, damage the school climate as well as the conditions necessary for learning [8].

Recent studies have explored the relationship between academic performance and the student’s perception of school coexistence from a multidimensional approach in which the negative impact that this has on the levels of indiscipline, aggressiveness, victimization and apathy from teachers, and academic performance is highlighted [9]. Moreover, school coexistence has been detected as a factor that greatly affects learning, in direct relationship with the type of interactions that prevail in schools, dramatically impacting academic performance [10]. Thus, school coexistence can positively or negatively affect academic performance, not only in terms of learning processes, but also in several areas of personal development such as the perception of subjective well-being [11] or self-esteem [12]. In this context, three dimensions are necessary for forming well-rounded students: cognitive, procedural, and attitudinal [13]. Apparently, the type of school coexistence could have a positive or negative impact on the three dimensions raised. This study will focus specifically on the effects it could have on school performance.

Several studies, particularly in Chile, have evidenced that a significant number of students perceive negative behaviors in their schools, that verbal violence is linked to ridicule, insults, and threats, and that this violence can be manifested explicitly or implicitly [13,14]. For instance, a study carried out in different schools detected that most behaviors associated with bullying were exhibited by boys in middle to low-class public schools [15]. Moreover, in a study, 9% of students reported being victims of verbal, social, or physical harassment, either directly or through social media [15] and in another study, 15.3% experienced bullying and victimization [16].

International evidence has shown that the students who report being victims of bullying and harassment also present negative educational patterns, low self-perception of academic efficacy, and poor academic performance [9,17,18]. In this scenario, this study seeks to highlight the importance of studying and analyzing school coexistence within the school community, especially due to its effects on the students’ physical, psychological, and emotional health. In addition, this study contributes to the incipient national evidence that has analyzed the impact of school coexistence on school performance [9,19]. The study’s objective was to identify the school coexistence indices and analyze the differences in academic performance according to school coexistence indices in students from public schools in the province of Biobío, Chile. Based on the existing evidence, we hypothesize that students perceive situations of violence and aggressiveness within the school and unsafe school areas. In addition, it was also hypothesized that students who perceive poor school coexistence have poorer academic performance compared to those who perceive good school coexistence.

## 2. Materials and Methods

### 2.1. Study Design

A nonexperimental, analytic, cross-sectional study was conducted following the STROBE guidelines [20]. For this study, data from the 2018 school coexistence indices and academic performance survey in the Biobío Province were used.

### 2.2. Population and Sample

All students from the fifth to eighth grades in all public schools (five public schools) in a district from the Biobío Province, Chile were invited to participate (*n* = 3.857). A probabilistic sample stratified by the gender of 797 Chilean students (12 ± 1.3) was included. A total of 67 students were excluded due to not attending on the day of the measurements or because their parents did not consent. The final sample consisted of 730 students (53.8% boys and 46.2% girls; 12 ± 1.2 years) from five public schools from the Biobio Province, Chile. To calculate the sample size, an error percentage of 5% and 95% confidence were considered.

### 2.3. Procedure

An alliance between the inter-university research team and the district’s education department (DAEM) was formed. Once the approvals of the ethics committee and the municipality’s authorization were obtained, the data collection was planned in conjunction with the school boards. After that, the teachers who applied the instruments were trained to reduce inter-rater bias. Consequently, the data collection was carried out on one day during the established class time (October 2019). The families, teachers, and the school board were informed about the purposes of the study. Then, the parents agreed to take part in the study by signing the informed consent. The study was carried out following the ethical, legal, and regulatory frameworks for human subject research. The study protocol was approved by the Bioethics Committee of the Universidad de Concepción, Chile (protocol code and approved date RZL-Abril/2018), and all procedures were carried out according to the Declaration of Helsinki for research with human subjects.

### 2.4. Outcomes and Instruments of Measurement

*Academic performance*: The students’ academic performance was measured through their accumulated final grades (AFG) and the grade point average (GPA). Accumulated final grades in the subjects of language (Spanish), mathematics, and physical education and health as well as the GPA reached during the first academic semester were considered. The marks ranged from 1.0 to 7.0, where 4.0 was the minimum passing mark. No differences in academic requirements were considered since all schools are public and are regulated by the same curriculum provided by the Ministry of Education. GPAs have been widely used in studies focusing on academic performance in Chilean educational centers [21].

*School coexistence indices*: For the school coexistence indices, a self-designed and validated instrument through expert judgement was used. This survey was adapted from other instruments validated and translated into Spanish that assess school climate and coexistence in elementary schools [22,23,24]. Our instrument intends to evaluate the students’ perception of concepts related to school coexistence, specifically (1) the quality of interpersonal relationships within the different school bodies. A 5-point Likert scale was used to determine the quality of interpersonal relationships. The scale went from very bad to very good to determine the quality of interpersonal relationships using the following classification: very bad, bad, neither good nor bad, good, and very good. (2) Perception of levels of violence and aggressiveness among peers from a gender perspective. A 5-point Likert scale was used to determine the perception of violence and aggressiveness from a gender perspective. The following classifications was used: strongly disagree, disagree, neither agree nor disagree, agree, and strongly agree. (3) Perception of safety and unsafety levels in different school areas. A 5-point Likert scale was used to classify the students’ responses as very unsafe, unsafe, neither safe nor unsafe, safe, and very safe. Our instrument has been used in Chile to build a diagnostic base in terms of the behaviors related to school coexistence in public schools, highlighting the importance of strengthening quantifiable elements to design appropriate school coexistence and management plans in the school community.

*Socio-educational data*: Additionally, the students’ age, sex, and grade as well as their participation or non-participation in the school integration programs according to their AFG and GPA were reported.

### 2.5. Statistical Analysis

The qualitative data were represented through absolute and percentage frequency n(%), while the quantitative data were represented through the mean (M) and standard deviation (SD). The distribution of data was analyzed through a Kolmogorov–Smirnoff test, which showed that all variables had a normal distribution. Moreover, the equality of variances was evidenced through Levene’s test, consequently, parametric statistics were employed in all analyses. Furthermore, to determine the differences in AFG and the GPA according to the socio-educational characteristics and school coexistences indices, a one-way ANOVA was used. Subsequently, if intragroup differences were found, a post hoc Bonferroni’s test was employed, aimed at identifying any differences between groups. Effect size (ES) was determined by eta-square (η^2^). An eta-square of 0 means no influence, while 1 indicates a full dependency. η^2^: 0.00 < 0.01 Negligible, η^2^ 0.01 < 0.06 Small, η^2^ 0.06 < 0.14 Medium, η^2^ 0.14 < 1.00 Large. The data were analyzed through the statistical software SPSS 19.0 (IBM SPSS statistics, Chicago, IL, USA). The level of significance used was *p* value ≤ 0.05.

## 3. Results

### 3.1. Socio-Educational Characteristics According to Accumulated Final Grades (AFG) and the Grade Point Average (GPA)

It was observed that most students were male (56.1%), aged between 12 and 13 years old (52.5%), and did not participate in the school integration program (80.8%). It was observed that the girls had a better AFG in language and a better GPA compared to the boys. Schoolchildren 10–11 years of age had better AFG in math and physical education, and a better GPA compared to older schoolchildren. Regarding grades, it was observed that mostly fifth-grade students had better AFG and GPA than older-grade students. In addition, it was evidenced that the schoolchildren who did not participate in the school integration program had better AFG and GPA compared to the schoolchildren who did participate in the school integration program (Table 1).

### 3.2. Description of School Coexistence Indices

Table 2 shows the indices of school coexistence according to the quality of interpersonal relationships between different school bodies, the perception of violence and aggressiveness among peers from a gender perspective, and the perception of safety levels in different school areas. It was observed that most of the students characterized the interpersonal relationships between the different school bodies as good or very good. Moreover, 73.6% described their relationships among peers as good or very good, and 76.5% reported that the relationships between students and head teachers were good or very good. Nevertheless, a large percentage of students agreed or strongly agreed with the statement expressing that stronger students were violent toward weaker students (40.9%) and that boys were violent toward one another (41.3%).

Additionally, when asked about safety in different school areas, most students expressed that all areas were safe or very safe. However, 20.5% of the students indicated that the restrooms were an unsafe or very unsafe space (Table 2).

### 3.3. AFGs and GPA According to Quality of Interpersonal Relationships between School Bodies

Table 3 depicts the academic performance according to the quality of interpersonal relationships between school bodies. Through a one-way ANOVA, differences in the accumulated final grades in math, language (Spanish), physical education, and GPA, according to the quality of interpersonal relationships between school bodies, were observed. A deeper analysis among peers (post hoc Bonferroni tests) pointed out that students who perceived bad relationships between students and the head teacher had lower AFG in math, language (Spanish), physical education and health, and GPA (*p* < 0.0001). It was observed that students who perceived very bad relationships between students and teachers obtained lower accumulated final grades in physical education and health (*p* < 0.0001) and a lower GPA (*p* = 0.002) compared to students who identified a good or very good relationship between students and teachers. Moreover, it was pinpointed that students who classified the relationship between the school headmaster and the students as very bad also had lower AFG in language (Spanish) and GPA than the students who considered said relationship as good or very good (*p* < 0.0001). Additionally, it was recognized that the students who stated that the relationship between students and inspectors was bad had a lower AFG in physical education and health and GPA in comparison with the students who identified said relationship as good or very good (*p* = 0.009). Furthermore, the students who reported a bad or very bad relationship between parents or guardians and the school showed lower results in math (*p* = 0.023) and lower GPA (*p* = 0.06) when compared to students who had a positive perception of said relationship.

### 3.4. AFG and GPA According to Perception of Violence and Aggressiveness among Peers from a Gender Perspective

Table 4 portrays the academic performance according to the perception of violence and aggressiveness among peers from a gender perspective. Through a one-way ANOVA, it was possible to observe differences in the AFG in physical education and health (*p* = 0.007) and GPA (*p* = 0.027), according to the perception of violence and aggressiveness among peers and differences in the AFG in math (*p* = 0.006), language (*p* < 0.0001), and GPA (*p* < 0.0001) among girls. A deeper analysis between groups (post hoc Bonferroni test) showed that students who strongly agreed that fights between students and between girls were frequent also presented lower AFG in physical education and health, language, and GPA when compared to every other group.

### 3.5. AFGs and GPA According to Perception of Levels of Safety and Unsafety in Different School Areas

Table 5 shows the academic performance according to the perception of levels of safety and unsafety in different areas of the school. A one-way ANOVA exposed differences in the final accumulated grades in math (*p* < 0.0001), language (Spanish) (*p* = 0.004), and GPA (*p* < 0.0001) when the classroom was perceived as an unsafe space. Differences in GPA were present when the halls (*p* = 0.019) and the canteen (*p* = 0.015) were perceived as unsafe. Moreover, differences in the accumulated final grades in math (*p* = 0.032) and GPA (*p* = 0.028) were noted when the fields and gyms were perceived as unsafe. In general terms, after conducting a deeper analysis between pairs of groups (Bonferroni post hoc test), it was evidenced that the students who perceived the classroom, halls, canteen, and gyms and fields as unsafe spaces obtained lower accumulated final grades in math and language as well as lower GPA in comparison with the rest of the students.

## 4. Discussion

The main results of this study reflect that a great percentage of schoolchildren agreed or strongly agreed that stronger students were violent toward weaker students and that boys were violent toward one another, respectively. Furthermore, the school areas most classified as unsafe or very unsafe by the students were the restrooms, followed by the playgrounds and the gym and fields. Furthermore, the students who classified the relationships within the school community as bad, or very bad, were the ones with lower AFG in subjects such as math, language in Spanish, and physical education and health as well as GPA. The students who perceived violence and frequent aggressiveness among peers and girls and higher levels of insecurity in different school areas presented poorer academic performance.

The results of this study show that the restrooms were identified as the least safe space in the school (20.5%). Studies in the area have shown similar results. An example of this was shown by Chávez Romo et al. (2017), who reported that teenagers encountered more violence in the halls, restrooms, and places surrounding the school, since these places are not usually under surveillance, and therefore, more unsafe. The results of this study point to the restrooms as the most unsafe space, in accordance with other studies along the same line [25]. Similarly, a study carried out in 2009, in which 282 polytechnic schools from the city of Valencia participated, distinguished that these aggressions occurred more often in the classroom (30.9%), however, the restrooms were also included in the topography of violent acts [26].

Respecting the association between school coexistence indices and academic performance, this study observed that the students who described the relationships between students and teachers or head teachers as bad or very bad exhibited a poorer academic performance when compared to the students who described said relationships as good or very good. Similarly, previous studies have highlighted school coexistence as a factor that greatly influences efficacy and educational attainment, since the students’ perceptions about their school climate significantly affect their academic performance [17,18,27]. Remarkably, relationships among students and between students and teachers were the ones that affected academic success the most [6]. Thus, similar lines of research have concluded that a paramount element in the success of 447 elementary school students from 32 subsidized and public schools in Santiago, in the Metropolitan region of Chile, is the head teacher, who is a significant figure [28]. Another finding of this study was that students with a bad perception of the relationships between students and inspectors obtained lower accumulated final grades in physical education in comparison with the students who described said relationship as good or very good (*p* = 0.008).

Research in this area has shown that students are affected by the quality of their relationships with inspectors or headmasters when punishment regarding discipline is administered, explicitly when students are taken out of their classrooms for a significant period, without addressing the issues with the students involved [29], instead excluding disruptive behavior through punishment [30]. Moreover, the study observed that the students who perceived recurrent violence and aggressiveness among boys or among girls portrayed poorer academic performance. This point is relevant since other studies have pointed to a more positive school coexistence among girls and that gender is a critical factor in school coexistence [15], although it is not a predictor of school violence [31]. Similar to our results, studies carried out by UNESCO and CEPAL in Latin America have reported that violence, verbal harassment, and arguments between students and teachers are transversal and undesired in the school system, and that they affect academic performance [32]. Furthermore, when students felt unsafe in the classroom, the halls, canteen, and gyms and fields, they showed poorer academic performance, in accordance with studies related to violence and bullying, which point to these places being the most common places for these types of acts. Among the causes of these occurrences, it is possible to find a lack of surveillance during recess, an increased density of population in the same area, differences in the age or interests of the students who get together, and disputes over spaces [33], which can undoubtedly cause a shift of attention from the academic aspects, and influence school performance.

Nevertheless, it is important to consider that this study also had certain limitations. First, a self-elaborated instrument based on validated instruments to measure school coexistence indices was applied [22,23,24]. This limits the possibility of contrasting the results to similar populations that measured similar school coexistence indices or applied standardized instruments. Moreover, the participants were asked to fill out the form, therefore, their answers were subject to their moods, understanding of the questions, and even the level of maturity to understand the relevance of their answers. To cope with this, the teachers in charge of applying the instruments were trained to answer any questions the students may have had. This data collection method has been widely used in the existing literature since it allows for great amounts of information from the students to be gathered quickly.

### 4.1. Strengths of This Study

This study contributes to the theoretical framework related to the possible negative effects of poor school coexistence, particularly regarding its effects on school performance. These results show that there was a high percentage of students who perceived a bad school environment in their educational center. In addition, it was observed that when low indicators of school coexistence were perceived, students had lower school performance. These results are in line with evidence suggesting significant indirect effects of bullying on achievement via psychological difficulties [34]. Hence, it is important for school authorities to act in the early detection of students who generate conflict and a bad school coexistence, delving deep into their motivations, the unsafe spaces in the school, and the intensity and frequency of the physical and psychological aggression toward weaker students, in order to generate actions that support the resolution of this conflict, which greatly affects the academic and general well-being of the students.

Furthermore, this study contributes to the knowledge about the effects of school coexistence on academic performance in Chilean public schools, which has not received the required scientific attention for such a relevant topic. Information regarding the students’ perceptions on different school coexistence indices was gathered and compared with their marks. In this respect, other studies in Latin America have been carried out aimed at improving the working conditions to boost school performance. For instance, a doctoral thesis developed in Perú in 2008, named *School Climate and Academic Performance*, and carried out with twelfth grade students, achieved positive results in academic performance by developing the students’ potential, skills, abilities, values, and attitudes [35]. Considering the research, these results are the most up-to-date evidence in Latin America that accounts for the direct relationship that school coexistence has on academic performance. In addition, it contributes to the incipient national evidence on the negative effects of poor school coexistence on the performance of public-school students.

### 4.2. Contributions, Practical Implications, and Future Lines of Research

The results of this study present evidence that allows educational centers to generate spaces for reflection and action regarding how school coexistence is being treated, and the impact it produces on the learning and quality of life of the students as well as the violence that persists within the school community, consolidating a coexistence culture. More research on the area is needed to understand this topic more deeply by using standardized instruments to gather information regarding concrete facts and actions pointing toward timely intervention, when necessary.

It is very important that future research includes students with permanent special educational needs to identify the type of support that is required, since the school integration program in Chile includes mostly schoolchildren with transitory educational needs. School organization is paramount and involves developing strategies that contribute to improve school coexistence and climate. Schools need to transform into a safe space, where the students who will build the future can experience appropriate growth.

## 5. Conclusions

In conclusion, students perceived violence and aggressiveness among themselves, and the spaces considered as the most unsafe in schools were ascertained (restrooms, playgrounds, and the gym and fields). Furthermore, it was evidenced that the students who perceived poor indices of school coexistence, especially regarding relationships between different school bodies, violence and aggressiveness among peers, and unsafe spaces, exhibited poorer academic performance than the students who perceived better indices of school coexistence.

These elements highlight the relevance of the effects that violence and aggression have in school spaces, and the need for appropriate management of these situations. It is paramount that schools address emotional and behavioral issues caused by bullying in order to improve the overall educational experience of a child [34].

Therefore, it is essential to promote the evaluation of the school climate, in addition to promoting positive practices and providing relevant support, to make improvements in the quality of life of the entire educational community [36].

## Figures and Tables

**Table 1 behavsci-13-00154-t001:** Socio-educational characteristics according to the AFG and GPA.

			Accumulated Final Grades							
			Mathematics		Language (Spanish)		Physical Education and Health		GPA	
Variables	*n*	(%)	Mean	*p* Value	Mean	*p* Value	Mean	*p* Value	Mean	*p* Value
Sex										
*Boys*	393	53.8	5.2 (1.0)	0.892	5.0 (0.8)	0.006	6.4 (0.5)	0.227	5.6 (0.6)	0.001
*Girls*	337	46.2	5.2 (1.0)		5.2 (0.8)		6.4 (0.5)		5.7 (0.6)	
Age										
*10–11 years*	261	35.8	5.3 (1.1) a	0.000	5.2 (0.9)	0.057	6.5 (0.4) a	0.000	5.8 (0.6) a	0.000
*12–13 years*	383	52.5	5.1 (1.0) b		5.1 (0.8)		6.4 (0.5) b		5.6 (0.6) b	
*14–15 years*	86	11.7	4.9 (0.9) b		5.0 (0.8)		6.1 (0.8) c		5.4 (0.6) c	
Grade										
*Fifth*	162	22.2	5.3 ± 1.0 (a)	0.000	5.3 (0.9) a	0.001	6.4 (0.5) b	0.000	5.7 (0.6) a	0.000
*Sixth*	187	25.6	5.3 ± 1.1 (a)		4.9 (0.8) b		6.6 (0.4) a		5.7 (0.6) a	
*Seventh*	197	27	4.9 ± 0.9 (b)		5.1 (0.8) a		6.3 (0.5) bc		5.6 (0.6) b	
*Eighth*	184	25.2	5.1 ± 1.0 (a)		5.1 (0.7) a		6.2 (0.7) c		5.5 (0.6) b	
School integration program										
*Yes*	140	19.2	4.8 (0.7)	0.000	4.8 (0.7)	0.000	6.2 (0.7)	0.000	5.3 (0.6)	0.000
*No*	590	80.8	5.3 (1.0)		5.2 (0.8)		6.4 (0.5)		5.7 (0.6)	

Caption: The qualitative data were represented through absolute and percentage frequency and the quantitative data were represented through the mean ± standard deviation. The statistical analysis was carried out through a one-factor ANOVA. In the same row (a–c), marks with different symbols indicate significant differences between groups through the Bonferroni post hoc test. Differences were significant with a value *p* < 0.05. *n* = 730. Source: Own elaboration.

**Table 2 behavsci-13-00154-t002:** School coexistence indices such as the quality of interpersonal relationships between school bodies, the perception of violence and aggressiveness from a gender perspective, and the perception of levels of safety and unsafety in different school areas.

Quality of Interpersonal Relationships between School Bodies										
	Very Bad		Bad		Neither Good nor Bad		Good		Very Good	
Students and head teacher	6	(0.8%)	14	(1.9%)	152	(20%)	331	(45.3%)	227	(31.1%)
Students and teachers	5	(0.7%)	26	(3.6%)	258	(35.3%)	339	(46.4%)	102	(14.0%)
Students and headmaster and administrative personnel	8	(1.1%)	31	(4.2%)	266	(36.4%)	285	(39.0%)	140	(19.2%)
Students and inspectors	13	(1.8%)	32	(4,4%)	219	(30%)	316	(43.3%)	150	(20.5%)
Parents/guardians and school	7	(1%)	12	(1.6%)	212	(29.0%)	324	(44.4%)	175	(24.0%)
Among students	11	(1.5%)	20	(2.7%)	163	(22.3%)	275	(37.7%)	261	(35.8%)
Perception of violence and aggressiveness from a gender perspective										
	Strongly disagree		Disagree		Neither agree nor disagree		Agree		Strongly agree	
*There are fights between students often*	96	(13.2%)	160	(21.9%)	293	(40.1%)	142	(19.5%)	39	(5.3%)
*Stronger students are violent toward weaker students*	123	(16.8%)	124	(17.0%)	184	(25.2%)	209	(28.6%)	90	(12.3%)
*Boys are violent and aggressive toward girls*	226	(31.0%)	194	(26.6%)	196	(26.8%)	85	(11.6%)	29	(4.0%)
*Girls are violent and aggressive toward boys*	164	(22.5%)	171	(23.4%)	217	(29.7%)	115	(15.8%)	62	(8.6%)
*Boys are violent toward one another*	78	(10.7%)	116	(15.9%)	235	(32.2%)	221	(30.3%)	80	(11.0%)
*Girls are violent toward one another*	166	(22.7%)	186	(25.5%)	233	(31.9%)	103	(14.1%)	42	(5.8%)
Perception of levels of safety and unsafety in different school areas										
	Very unsafe		Unsafe		Neither safe nor unsafe		Safe		Very safe	
*Entrances and exits*	20	(2.7%)	36	(4.9%)	175	(24.0%)	298	(40.8%)	201	(27.6%)
*Classrooms*	8	(1.1%)	25	(3.4%)	94	(12.9%)	311	(42.6%)	292	(40.0%)
*Halls*	13	(1.8%)	40	(5.5%)	190	(26.0%)	306	(41.9%)	181	(24.8%)
*Playgrounds*	12	(1.6%)	61	(8.4%)	200	(27.4%)	267	(36.6%)	190	(26.2%)
*Restrooms*	46	(6.3%)	103	(14.1%)	221	(30.3%)	223	(30.5%)	137	(18.9%)
*Canteen*	24	(3.3%)	29	(4%)	158	(21.6%)	299	(41.0%)	220	(30.2%)
*Gyms and fields*	20	(2.7%)	50	(6.8%)	136	(18.6%)	270	(37%)	254	(34.9%)

Caption: The qualitative data were represented through absolute and percentage frequency. Source: Own elaboration.

**Table 3 behavsci-13-00154-t003:** AFG and GPAs according to the quality of the interpersonal relationships between school bodies.

Academic Performance	Very Bad	Bad	Neither Good nor Bad	Good	Very Good	ES	One-Way ANOVA
	M (SD)	M (SD)	M (SD)	M (SD)	M (SD)	η^2^	*p*-Value
Students and head teacher							
Math (1–7)	4.1 (1.3) a	4.7 (1.0) ab	5.0 (1.0) ab	5.2 (1.0) b	5.3 (1.0) b	0.030	<0.0001
Language (Spanish) (1–7)	4.3 (0.5) a	4.6 (0.8) a	5.0 (0.8) a	5.1 (0.8) ab	5.3 (0.8) b	0.033	<0.0001
PE&H (1–7)	5.5 (1.1) a	6.5 (0.4) b	6.4 (0.5) b	6.4 (0.6) b	6.4 (0.5) b	0.027	<0.0001
GPA (1–7)	4.6 (0.6) a	5.3 (0.7) a	5.5 (0.5) b	5.6 (0.6) bc	5.8 (0.6) c	0.064	<0.0001
Students and teachers							
Math (1–7)	4.7 (1.4)	5.0 (1.0)	5.1 (1.0)	5.2 (1.0)	5.4 (1.1)	0.011	0.097
Language (Spanish) (1–7)	4.8 (0.4)	5.0 (0.8)	5.0 (0.8)	5.1 (0.8)	5.3 (1.0)	0.013	0.055
PE&H (1–7)	5.5 (0.6) a	6.1 (0.9) ab	6.4 (0.5) b	6.4 (0.5) b	6.4 (0.6) b	0.027	<0.0001
GPA (1–7)	5.1 (0.8) ab	5.4 (0.7) ab	5.6 (0.6) a	5.7 (0.6) ab	5.8 (0.7) b	0.023	0.002
Students and headmaster							
Math (1–7)	5.3 (0.5) ab	4.9 (0.9) a	5.1 (1.0) a	5.2 (1.0) ab	5.4 (1.0) b	0.019	0.007
Language (Spanish) (1–7)	5.1 (0.5) ab	5.1 (0.8) a	5.0 (0.79) a	5.0 (0.8) ab	5.4 (0.9) b	0.029	<0.0001
PE&H (1–7)	6.2 (0.8)	6.2 (0.7)	6.4 (0.5)	6.4 (0.6)	6.4 (0.6)	0.009	0.170
GPA (1–7)	5.5 (0.3) ab	5.5 (0.6) ab	5.6 (0.6) a	5.6 (0.6) ab	5.8 (0.6) b	0.029	<0.0001
Students and inspector							
Math (1–7)	5.3 (0.6)	5.0 (0.8)	5.1 (1.0)	5.2 (1.0)	5.3 (1.0)	0.006	0.362
Language (Spanish) (1–7)	5.3 (0.6)	4.9 (0.6)	5.0 (0.8)	5.2 (0.8)	5.2 (0.9)	0.005	0.416
PE&H (1–7)	6.3 (0.6) ab	6.1 (0.6) a	6.4 (0.5) b	6.4 (0.5) b	6.4 (0.5) b	0.019	0.009
GPA (1–7)	5.6 (0.6) ab	5.4 (0.5) ab	5.6 (0.6) a	5.7 (0.6) ab	5.7 (0.6) b	0.018	0.009
Parents/guardians and school							
Math (1–7)	4.4 (1.3)a	4.6 (0.3) ab	5.1 (1.0) ab	5.2 (1.1) ab	5.3 (1.0) b	0.015	0.023
Language (Spanish) (1–7)	4.7 (0.6)	4.8 (0.4)	5.2 (0.8)	5.1 (0.8)	5.1 (0.8)	0.008	0.186
PE&H (1–7)	6.5 (0.7)	6.1 (0.7)	6.4 (0.5)	6.4 (0.6)	6.4 (0.5)	0.008	0.232
GPA (1–7)	5.3 (0.8) ab	5.2 (0.3) a	5.6 (0.5) ab	5.6 (0.6) ab	5.7 (0.6) b	0.020	0.006
Between students							
Math (1–7)	5.0 (1.1)	5.1 (1.0)	5.2 (1.0)	5.1 (1.0)	5.2 (1.0)	0.006	0.398
Language (Spanish) (1–7)	4.8 (0.7)	5.1 (0.8)	5.1 (0.9)	5.1 (0.8)	5.1 (0.8)	0.003	0.768
PE&H (1–7)	6.2 (0.8) ab	6.1 (0.8) a	6.4 (0.5) ab	6.4 (0.6) ab	6.4 (0.5) b	0.015	0.023
GPA (1–7)	5.3 (0.7)	5.7 (0.6)	5.7 (0.6)	5.6 (0.6)	5.6 (0.6)	0.005	0.486

Caption: The statistical analysis was carried out through a one-factor ANOVA. In the same row (a,b,c), marks with different symbols indicate significant differences between groups through the Bonferroni post hoc test. Differences were significant with a value *p* < 0.05. *n* = 730. PE&H: Physical Activity and Health. Source: Own elaboration.

**Table 4 behavsci-13-00154-t004:** AFG and GPA according to the perception of violence and aggressiveness among peers from a gender perspective.

Academic Performance	Strongly Disagree	Disagree	Neither Agree nor Disagree	Agree	Strongly Agree	ES	One-Way ANOVA
	M (SD)	M (SD)	M (SD)	M (SD)	M (SD)	η^2^	*p*-Value
Fights among peers are frequent							
Math (1–7)	5.2 (1.0)	5.2 (1.0)	5.2 (1.0)	5.1 (1.0)	4.8 (0.9)	0.010	0.135
Language (Spanish) (1–7)	5.1 (0.8)	5.2 (0.8)	5.1 (0.8)	5.1 (0.9)	5.0 (0.9)	0.003	0.683
PE&H (1–7)	6.4 (0.6) ab	6.4 (0.5) ab	6.4 (0.5) a	6.4 (0.6) a	6.1 (0.7) b	0.019	0.007
GPA (1–7)	5.7 (0.6) a	5.6 (0.6) ab	5.7 (0.6) a	5.6 (0.7) ab	5.4 (0.6) b	0.015	0.027
Stronger students are violent toward weaker students							
Math (1–7)	5.3 (1.0)	5.1 (1.1)	5.2 (1.0)	5.1 (1.0)	5.0 (0.9)	0.006	0.384
Language (Spanish) (1–7)	5.2 (0.8)	5.1 (0.8)	5.1 (0.8)	5.1 (0.8)	5.0 (0.9)	0.003	0.670
PE&H (1–7)	6.3 (0.6)	6.4 (0.6)	6.4 (0.6)	6.4 (0.5)	6.5 (0.5)	0.004	0.574
GPA (1–7)	5.7 (0.6)	5.6 (0.6)	5.6 (0.6)	5.7 (0.6)	5.6 (0.6)	0.003	0.769
Boys are violent and aggressive toward girls							
Math (1–7)	5.2 (1.0)	5.2 (1.0)	5.2 (1.0)	4.9 (1.0)	5.1 (1.0)	0.010	0.131
Language (Spanish) (1–7)	5.1 (0.8)	5.2 (0.8)	5.2 (0.8)	4.9 (0.8)	4.9 (0.9)	0.011	0.086
PE&H (1–7)	6.4 (0.5)	6.4 (0.6)	6.4 (0.5)	6.3 (0.6)	6.3 (0.7)	0.010	0.131
GPA	5.7 (0.6)	5.7 (0.6)	5.7 (0.6)	5.5 (0.6)	5.6 (0.6)	0.008	0.208
Girls are violent and aggressive towards boys							
Math (1–7)	5.1 (1.0)	5.2 (1.1)	5.2 (1.0)	5.1 (1.0)	5.1 (0.9)	0.003	0.749
Language (Spanish) (1–7)	5.2 (0.8)	5.1 (0.8)	5.1 (0.8)	5.0 (0.8)	4.9 (0.7)	0.009	0.158
PE&H (1–7)	6.4 (0.5)	6.4 (0.5)	6.4 (0.5)	6.3 (0.5)	6.4 (0.6)	0.006	0.326
GPA (1–7)	5.7 (0.6)	5.7 (0.6)	5.6 (0.6)	5.5 (0.6)	5.6 (0.6)	0.008	0.230
Boys are violent and aggressive toward one another							
Math (1–7)	5.2 (1.0)	5.1 (1.0)	5.2 (1.0)	5.2 (1.0)	5.0 (1.0)	0.004	0.513
Language (Spanish) (1–7)	5.1 (0.8)	5.2 (0.8)	5.1 (0.8)	5.2 (0.8)	4.9 (0.9)	0.007	0.257
PE&H (1–7)	6.4 (0.6)	6.4 (0.6)	6.4 (0.5)	6.4 (0.6)	6.4 (0.5)	0.001	0.962
GPA (1–7)	5.7 (0.6)	5.6 (0.7)	5.7 (0.6)	5.6 (0.6)	5.5 (0.6)	0.005	0.471
Girls are violent toward one another							
Math (1–7)	5.2 (1.0) ab	5.1 (1.1) ab	5.3 (1.0) a	5.0 (1.0) b	4.8 (0.9) b	0.020	0.006
Language (Spanish) (1–7)	5.2 (0.8) a	5.1 (0.8) ab	5.2 (0.8) a	4.9 (0.8) b	4.7 (0.9) b	0.036	<0.0001
PE&H (1–7)	6.4 (0.6)	6.4 (0.5)	6.4 (0.5)	6.4 (0.6)	6.2 (0.6)	0.006	0.346
GPA (1–7)	5.7 (0.6) a	5.6 (0.6) ab	5.7 (0.6) a	5.5 (0.6) b	5.4 (0.7) b	0.030	<0.0001

Caption: The statistical analysis was carried out through a one-factor ANOVA. In the same row (a,b), marks with different symbols indicate significant differences between groups through the Bonferroni post hoc test. Differences were significant with a value *p* < 0.05. *n* = 730. PE&H: Physical Activity and Health. Source: Own elaboration.

**Table 5 behavsci-13-00154-t005:** Accumulated final grades and GPAs according to the perception of the levels of safety and unsafety in different school areas.

Academic Performance	Very Unsafe	Unsafe	Neither Safe nor Unsafe	Safe	Very Safe	ES	One-Way ANOVA
	M (SD)	M (SD)	M (SD)	M (SD)	M (SD)	η^2^	*p*-Value
Entrances and exits							
Math (1–7)	4.9 (0.8)	5.0 (0.9)	5.2 (1.0)	5.2 (1.0)	5.2 (1.1)	0.004	0.545
Language (Spanish) (1–7)	5.1 (0.8)	5.0 (0.7)	5.1 (0.8)	5.1 (0.8)	5.2 (0.9)	0.004	0.580
PE&H (1–7)	6.3 (0.6)	6.3 (0.7)	6.4 (0.5)	6.4 (0.6)	6.4 (0.6)	0.003	0.653
GPA (1–7)	5.6 (0.6)	5.6 (0.5)	5.6 (0.6)	5.6 (0.6)	5.7 (0.6)	0.002	0.845
Classrooms							
Math (1–7)	4.5 (0.5) ab	5.0 (1.1) ab	4.8 (1.0) b	5.1 (1.0) ab	5.3 (1.0) a	0.034	<0.0001
Language (Spanish) (1–7)	4.4 (0.6) b	5.1 (0.9) b	4.9 (0.7) ab	5.1 (0.8) ab	5.2 (0.8) a	0.021	0.004
PE&H (1–7)	6.3 (0.6)	6.3 (0.62)	6.3 (0.5)	6.4 (0.6)	6.4 (0.5)	0.004	0.607
GPA (1–7)	4.9 (0.3) b	5.6 (0.7) a	5.5 (0.6) ab	5.6 (0.6) a	5.7 (0.6) a	0.042	<0.0001
Halls							
Math (1–7)	4.7 (0.6)	4.9 (0.9)	5.2 (1.1)	5.1 (1.0)	5.2 (1.0)	0.011	0.081
Language (Spanish) (1–7)	4.7 (0.6)	4.9 (0.8)	5.1 (0.8)	5.1 (0.8)	5.2 (0.8)	0.011	0.098
PE&H (1–7)	6.3 (0.6)	6.3 (0.6)	6.5 (0.4)	6.3 (0.6)	6.4 (0.5)	0.009	0.141
GPA (1–7)	5.2 (0.5) b	5.5 (0.6) ab	5.7 (0.6) ab	5.6 (0.6) ab	5.7 (0.6) a	0.016	0.019
Playgrounds							
Math (1–7)	4.6 (0.6)	5.2 (1.0)	5.2 (1.0)	5.2 (1.0)	5.2 (1.1)	0.006	0.360
Language (Spanish) (1–7)	4.5 (0.7)	5.1 (0.8)	5.1 (0.8)	5.1 (0.8)	5.2 (0.9)	0.012	0.074
PE&H (1–7)	6.3 (0.6)	6.4 (0.5)	6.4 (0.5)	6.3 (0.6)	6.4 (0.6)	0.005	0.459
GPA (1–7)	5.2 (0.6)	5.7 (0.6)	5.7 (0.6)	5.6 (0.6)	5.7 (0.6)	0.011	0.103
Restrooms							
Math (1–7)	5.2 (1.0)	5.1 (1.0)	5.2 (1.0)	5.1 (1.0)	5.3 (1.0)	0.005	0.414
Language (Spanish) (1–7)	5.0 (0.7)	5.1 (0.9)	5.1 (0.8)	5.1 (0.8)	5.2 (0.8)	0.004	0.567
PE&H (1–7)	6.4 (0.5)	6.3 (0.5)	6.5 (0.4)	6.3 (0.6)	6.4 (0.5)	0.012	0.076
GPA (1–7)	5.6 (0.6)	5.6 (0.6)	5.6 (0.6)	5.6 (0.6)	5.7 (0.5)	0.006	0.386
Canteen							
Math (1–7)	4.6 (0.8)	5.2 (0.9)	5.2 (1.0)	5.2 (1.0)	5.2 (1.0)	0.010	0.126
Language (Spanish) (1–7)	5.0 (0.8)	5.1 (0.8)	5.1 (0.8)	5.1 (0.8)	5.1 (0.8)	0.003	0.739
PE&H (1–7)	6.2 (0.7)	6.5 (0.4)	6.4 (0.5)	6.4 (0.5)	6.4 (0.6)	0.005	0.448
GPA (1–7)	5.3 (0.8) b	5.6 (0.6) ab	5.6 (0.6) a	5.6 (0.6) a	5.7 (0.6) a	0.017	0.015
Gyms and fields							
Math (1–7)	4.9 (0.8) ab	5.6 (0.9) a	5.1 (0.8) ab	5.1 (0.8) b	5.2 (1.0) ab	0.014	0.032
Language (Spanish) (1–7)	5.1 (0.6)	5.4 (0.8)	5.1 (0.8)	5.1 (0.8)	5.1 (0.9)	0.008	0.231
PE&H (1–7)	6.2 (0.6)	6.4 (0.5)	6.4 (0.5)	6.4 (0.6)	6.4 (0.5)	0.007	0.273
GPA (1–7)	5.5 (0.6) ab	5.9 (0.5) b	5.6 (0.6) a	5.6 (0.6) a	5.6 (0.6) a	0.015	0.028

Caption: The statistical analysis was carried out through one-factor ANOVA. In the same row (a,b), marks with different symbols indicate significant differences between groups through the Bonferroni post hoc test. Differences were significant with a value *p* < 0.05. *n* = 730. PE&H: Physical Activity and Health. Source: Own elaboration.

## Data Availability

Data will be made available upon request.

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
