# Peer review of "Academic Performance according to School Coexistence Indices in Students from Public Schools in the South of Chile"

_behavsci, 2023, doi:10.3390/bs13020154_

Round 1

Reviewer 1 Report

This is a good piece of work. Need just some minor corrections. 

1. Updated the references with the most recent ones (relevant). 

2. Make the key contribution clear

3. Copy-edit the manuscript

Good luck. 

Author Response

General comment: This is a good piece of work. Need just some minor corrections. 

Answer: Thank you very much for you comment. We have worked a lot for this manuscript.

Comment 1: Updated the references with the most recent ones (relevant). 

Answer 1: We appreciate the comment. New and updated references have been incorporated in the introduction and discussion sections.

Comment 2: Make the key contribution clear

Answer 2: The section on contributions and practical implications has been reviewed and clarified.

Comment 3: Copy-edit the manuscript

Answer 3: We appreciate the comment. The new version of the manuscript has been fully edited and reviewed by a native English speaker.

Reviewer 2 Report

Academic performance is a topic that has been widely studied at the international level. This paper analyzed the relationship between students’ academic performance and school coexistence indexes in Chile. In general, the statistical analyses are not comprehensive, and the language is not fluent. Please refer to the following suggestions to improve the paper.

Abstract:

1. The results lack the description of “the quality of interpersonal relationships between school bodies”. The results of students’ perception of violence and aggressiveness from a gender perspective and perception of levels of safety and unsafety in different school areas are suggested to be more concise.

2. “Moreover, the students perceived violence and frequent aggressiveness among peers, and girls and higher levels of insecurity in different school areas presented a poorer academic performance.” It is hard to understand this sentence. And compared with AFG and GPA according to the quality of interpersonal relationships between school bodies, this sentence only mentions “a poorer academic performance”, rather than detailed AFG and GPA.

3. The keyword “school” is vague and may not be the best option for the study.

Introduction:

4. The logic is confusing and how the authors get the conclusion that “three dimensions are necessary for forming well-rounded students: cognitive, procedural, and attitudinal” in that paragraph.

5. The literature review is too general and mentions that school coexistence can positively or negatively affect students’ academic performance. However, the authors divide school coexistence into three types of perception, which is also the basis to analyze how school coexistence influences students’ academic performance. Therefore, it is suggested that the introduction should review the impact of these three perceptions on academic performance.

6. What is the innovation of the study? The authors mention the negative effects of school coexistence on students in Chile and propose the objectives of the study. In this paragraph, it is suggested to add the limitations of the existing research and the innovation of this study.

Materials and Methods:

7. “Study design” is too brief. I suggest that this section be merged with the following part “population and sample”.

8. Outcomes and instruments of measurement. The specific content of the three perceptual dimensions of school coexistence needs further explanation. And the reliability of the instrument used is not reported.

Results

9. The title of 3.2. could be simplified as “Description of the school coexistence indexes”.

10. “Moreover, 73.6% described relationships among peers as good or very good, and 76.5% reported relationships between students and head teachers were good or very good.” The numbers are different from those in the table.

11. “Nevertheless, a big percentage of students agreed or strongly agreed with the statement expressing that stronger students were violent towards weaker students (41.2%) and that boys were violent towards one another (41.5%)” The numbers in the sentence are different from those in the abstract and table.

12. The section “3.4. AFG and GPA according to perception of violence and aggressiveness among peers from a gender perspective” lacks the description of students’ academic performance with strong agreement that “girls are violent towards one another”.

Discussion

11. “Regarding perception of cognitive functions, …” What is the role of this paragraph? The paper does not talk about having poor attention and getting nervous in classes at all.

12. In the section “4.1. Strengths and limitations”, the relevance of the first paragraph to this study is confusing. This section mentions the innovation of this study, which is suggested to emphasize in the introduction.

13. Deleting the title “4.1. Limitations” is preferable.

Author Response

Academic performance is a topic that has been widely studied at the international level. This paper analyzed the relationship between students’ academic performance and school coexistence indexes in Chile. In general, the statistical analyses are not comprehensive, and the language is not fluent. Please refer to the following suggestions to improve the paper.

Answer: We appreciate your comment. In this regard, we consider that the statistical analysis is adequate and consistent with the objectives, hypothesis and study design. However, the statistical analysis of the study is rewritten and clarified.

Regarding language fluency, this new version of the manuscript has been revised and edited. In our opinion, its fluency improved significantly.

Abstract:

  1. The results lack the description of “the quality of interpersonal relationships between school bodies”. The results of students’ perception of violence and aggressiveness from a gender perspective and perception of levels of safety and unsafety in different school areas are suggested to be more concise.
  2. “Moreover, the students perceived violence and frequent aggressiveness among peers, and girls and higher levels of insecurity in different school areas presented a poorer academic performance.” It is hard to understand this sentence. And compared with AFG and GPA according to the quality of interpersonal relationships between school bodies, this sentence only mentions “a poorer academic performance”, rather than detailed AFG and GPA.
  3. The keyword “school” is vague and may not be the best option for the study.

Answer: the abstract is corrected and clarified. Regarding keywords: "school" is removed, and other more relevant keywords are added.

Introduction:

  1. The logic is confusing and how the authors get the conclusion that “three dimensions are necessary for forming well-rounded students: cognitive, procedural, and attitudinal” in that paragraph.
  2. The literature review is too general and mentions that school coexistence can positively or negatively affect students’ academic performance. However, the authors divide school coexistence into three types of perception, which is also the basis to analyze how school coexistence influences students’ academic performance. Therefore, it is suggested that the introduction should review the impact of these three perceptions on academic performance.
  3. What is the innovation of the study? The authors mention the negative effects of school coexistence on students in Chile and propose the objectives of the study. In this paragraph, it is suggested to add the limitations of the existing research and the innovation of this study.

Answer: The indicated paragraph is revised and rewritten, references are added to strengthen the theoretical framework and the problem of the study. In addition, arguments are provided to better justify the study.

Materials and Methods:

  1. “Study design” is too brief. I suggest that this section be merged with the following part “population and sample”.
  2. Outcomes and instruments of measurement. The specific content of the three perceptual dimensions of school coexistence needs further explanation. And the reliability of the instrument used is not reported.

Answer: Information on the study design, and characteristics of the evaluation instruments were added.

Results

  1. The title of 3.2. could be simplified as “Description of the school coexistence indexes”.
  2. “Moreover, 73.6% described relationships among peers as good or very good, and 76.5% reported relationships between students and head teachers were good or very good.” The numbers are different from those in the table.

  1. “Nevertheless, a big percentage of students agreed or strongly agreed with the statement expressing that stronger students were violent towards weaker students (41.2%) and that boys were violent towards one another (41.5%)” The numbers in the sentence are different from those in the abstract and table.
  2. The section “3.4. AFG and GPA according to perception of violence and aggressiveness among peers from a gender perspective” lacks the description of students’ academic performance with strong agreement that “girls are violent towards one another”.

Answer: All reviewer comments are taken into consideration. The results section is corrected. Title 3.2 is shortened, the percentages in section 3.2 are corrected, the text that was missing in section 3.4 has been added.

Regarding comment 10, the percentages described do not appear explicitly in Table 2 because, as indicated in the text, they are the sum of two categories.

Discussion

  1. “Regarding perception of cognitive functions, …” What is the role of this paragraph? The paper does not talk about having poor attention and getting nervous in classes at all.
  2. In the section “4.1. Strengths and limitations”, the relevance of the first paragraph to this study is confusing. This section mentions the innovation of this study, which is suggested to emphasize in the introduction.

  1. Deleting the title “4.1. Limitations” is preferable.

Answer: The discussion has been completely revised. Misleading information has been removed, the strengths section has been improved, and the limitations heading has been removed.

Reviewer 3 Report

Although the theme is accurate and interesting, I recommend considerably reformulating the manuscript

Academic performance according to school coexistence indexes in students from public schools in south of Chile (behavsci-2191794)

Initial comment: In general, the manuscript seems to me an appropiate paper to be published with major revisions. The authors did not make a sufficiently consistent introduction. The analysis methods are simple but adequate. The discussion responded to a greater or lesser extent to the research problems.

However, I would like to make some clarifications that would improve the quality of the article.

Please answer each of the sections separately:

Abstract: The abstract is too long (277 words). I recommend synthesizing it according to the journals's guidlines. Furhtermore, I recommend sorting keywords alphabetically and followed by “;”.

1. Introduction: The introduction is not appropriate. The introduction section does not summarizes the objective of the study. I recommend expanding the theoretical framework. The hypotheses or research problems must be established at the end of the section.

2. Material and methods:

Study design: More detailed information is required

Population and sample: It must be indicated how the sample was selected for the investigation. Informed consent must be mentioned.

Outcomes and instruments of measurement: A detailed description of the questionnaires as well as Cronbach's Alpha should be mentioned.

Statistical Analysis: Statistical analysis should be written more clearly relating it to the research objectives or intended results.

3. Results: The statistical treatment is simple. I would introduce some causal relationship in the form of a structural equation model for contrast models that propose causal relationships between the study variables.

4. Discussion: The discussion is correct according to the information provided, although I still think that a better statistical treatment would considerably improve the manuscript. In addition, the discussion must be in line with the hypotheses raised.

5. Conclusions: Appropriate conclusions in line with the results and the discussion. The sections "Strenghts and limitations” must be separated.

Minor revisions:

a)     Informed Consent Statement must be specified (Line 422)

b)    Keywords must be numbered alphabetically

c)     Revise the whole bibliography

Final comment: I would like to strongly encourage authors to reformulate their manuscript with the changes made in this document.

Thank you very much.

Author Response

Although the theme is accurate and interesting, I recommend considerably reformulating the manuscript

Academic performance according to school coexistence indexes in students from public schools in south of Chile (behavsci-2191794)

Initial comment: In general, the manuscript seems to me an appropiate paper to be published with major revisions. The authors did not make a sufficiently consistent introduction. The analysis methods are simple but adequate. The discussion responded to a greater or lesser extent to the research problems.

However, I would like to make some clarifications that would improve the quality of the article.

Please answer each of the sections separately:

Abstract: The abstract is too long (277 words). I recommend synthesizing it according to the journals's guidlines. Furhtermore, I recommend sorting keywords alphabetically and followed by “;”.

Answer: The abstract has been shortened and the keywords have been arranged alphabetically.

  1. Introduction: The introduction is not appropriate. The introduction section does not summarizes the objective of the study. I recommend expanding the theoretical framework. The hypotheses or research problems must be established at the end of the section.

Answer: Corrections are made, the framework in the introduction is expanded, the objective is adjusted, and a research hypothesis is added. Now there is methodological coherence in the objective, hypothesis and statistical analysis.

  1. Material and methods:

Study design: More detailed information is required.

Population and sample: It must be indicated how the sample was selected for the investigation. Informed consent must be mentioned.

Outcomes and instruments of measurement: A detailed description of the questionnaires as well as Cronbach's Alpha should be mentioned.

Statistical Analysis: Statistical analysis should be written more clearly relating it to the research objectives or intended results.

Answer 2: Information on the study design, sample size, informed consent, characteristics of the evaluation instruments and statistical analysis was added.

  1. Results: The statistical treatment is simple. I would introduce some causal relationship in the form of a structural equation model for contrast models that propose causal relationships between the study variables.

Answer: The statistical analysis is coherent and consistent with the research objectives and hypotheses.

The comment is appreciated, but the statistical analysis suggested by the reviewer is not consistent with the objectives of the current study and could lead to a totally different study.

  1. Discussion: The discussion is correct according to the information provided, although I still think that a better statistical treatment would considerably improve the manuscript. In addition, the discussion must be in line with the hypotheses raised.

Answer: The discussion has been aligned to the objectives and hypotheses raised.

  1. Conclusions: Appropriate conclusions in line with the results and the discussion. The sections "Strenghts and limitations” must be separated.

Answer 5: thank you for you comment. the "strengths" and "limitations" sections were separated.

Minor revisions:

  1. a)     Informed Consent Statement must be specified (Line 422)
  2. b)    Keywords must be numbered alphabetically
  3. c)     Revise the whole bibliography

Answer: We appreciate the comments. Minor revisions were done.

Final comment: I would like to strongly encourage authors to reformulate their manuscript with the changes made in this document.

 Answer: The suggested changes have been made. An improved version of the manuscript was submitted.